# Background Level of Unstable Chromosome Aberrations in the Kazakhstan Population: A Human Biomonitoring Study

**DOI:** 10.3390/ijerph19148485

**Published:** 2022-07-12

**Authors:** Laura B. Kenzhina, Aigul N. Mamyrbayeva, Sergey N. Lukashenko, Zhanat A. Baigazinov, Dina B. Biyakhmetova, Andrey V. Panitskiy, Elena Polivkina, Fail F. Zhamaldinov, Clarice Patrono, Valentina Palma, Antonella Testa

**Affiliations:** 1Institute of Radiation Safety and Ecology NNC RK, Beibit Atom 2, Building 23, Kurchatov 071100, Kazakhstan; mamyrbayeva@nnc.kz (A.N.M.); zh.baigazinov@gmail.com (Z.A.B.); biyakhmetova95@mail.ru (D.B.B.); panitskiy@nnc.kz (A.V.P.); polivkina@nnc.kz (E.P.); zhamaldinov@nnc.kz (F.F.Z.); 2Institute of Radiology and Agroecology, Kievskoe Shosse 109 km, 249032 Obninsk, Kaluga Region, Russia; lukashenko.1962@mail.ru; 3National Agency for New Technologies, Energy and Sustainable Economic Development (ENEA), Casaccia Research Center, Via Anguillarese, 301, 00123 Rome, Italy; clarice.patrono@enea.it (C.P.); valentina.palma@enea.it (V.P.); antonella.testa@enea.it (A.T.)

**Keywords:** radiation-induced chromosome aberrations, Kazakhstan population, dicentrics, background frequency

## Abstract

Kazakhstan is known as a country with a complex radioecological situation resulting from different sources such as a natural radiation background, extensive activities of the industrial system of the former Soviet Union and a well-known testing of nuclear power weapons occurred in the Semipalatinsk Test Site (STS) area. The present study focuses on the assessment of the background of dicentric chromosomes in Kazakhstan’s population, which is the starting point in the dose assessment of irradiated people, since the baseline level of spontaneous dicentrics can vary significantly in different populations. In this context, aiming to determine the background frequency of chromosome aberrations in the population of Kazakhstan, considering the heterogeneity of natural radiation background levels of its large territory, a selection of 40 control subjects living in four cities of North, South, West and East Kazakhstan was performed. The cytogenetic study on the selected groups showed fairly low background frequency values of chromosome aberrations (0.84 ± 0.83 per 1000 cells), comparable with other data in the literature on general populations, reporting background frequency values between 0.54 and 2.99 per 1000 cells. The obtained results should be taken into account when constructing the dose–effect calibration curve used in cytogenetic biodosimetry, as a “zero” dose point, which will reduce the uncertainty in quantifying the individual absorbed dose in emergency radiological situations.

## 1. Introduction

Kazakhstan is known as a country with a complex radioecological situation resulting from different sources such as natural radiation background, extensive activities of the industrial system of the former Soviet Union and a well-known testing of nuclear power weapons.

Due to the specific geological structure, natural uranium-containing and hydrocarbon resource endowment and history of nuclear test sites as a raw material base for the military–industrial complex, Kazakhstan has accumulated extensive experience of radiation exposure. In particular, the Semipalatinsk region (STS) in eastern Kazakhstan was used for testing nuclear weapons for the Soviet army during the period 1949–1989. Furthermore, Kazakhstan has a unique uranium division of the “Ulba Metallurgical Plant” JSC, one of the world’s largest producers of nuclear fuel for nuclear power plants. In addition, East Kazakhstan owns a fuel processing plant and an international low-enriched uranium storage bank, which is a spent nuclear fuel storage facility. A significant part of the territory has been contaminated with natural and artificial radionuclides including some regions containing high levels of radon [1,2,3,4].

Mining and processing of minerals, in particular uranium ore and thorium, are the main priority in Kazakhstan’s economics [5]. The dozens of uranium deposits discovered in the territory of Kazakhstan are different in terms of formation conditions and practical value. Approximately 41% of the world’s reserves of uranium are concentrated in Kazakhstan, especially in the North and West regions with their uranium ore and thorium-containing provinces (i.e., Shu–Sarysu, Syrdarya and Shu–Ili provinces), where the most significant deposits of uranium and thorium and their daughter decay products are located [6,7]. There are plenty of companies located in different regions, engaged in the production of uranium, beryllium, tantalum and niobium items, holding a leading position in the supply to Europe, America, Canada and Asia.

For all these reasons, Kazakhstan’s current situation in terms of radiological safety provides a unique opportunity for a wide range of studies including biomonitoring studies and retrospective dosimetry investigations.

Several studies concerning radiation exposure rely heavily on the quantification of chromosome aberrations, such as dicentric chromosomes (Dic) and acentric fragments (Ace), in the peripheral blood lymphocytes (PBLs) of exposed and potentially exposed subjects. Dicentric chromosomes are specific radiation-induced aberrations occurring at a very low level in unirradiated persons (<0.001) [8] and increasing in a linear [9] or linear quadratic manner [10] after exposure to high- or low-LET radiation, respectively. Thus, in the context of the biological dosimetry, estimates of an absorbed whole-body dose can be determined by the dicentric frequency observed in PBLs [11].

Throughout the world, biodosimetry labs are pursuing various strategies to increase their capacity, establishing national or international biodosimetry networks of mutual assistance [12]. Several networks have already been established, e.g., the North American BD Network from Canada and The United States of America Cytogenetic Emergency Network (CEN) [13], the Latin American Biological Dosimetry Network (LBDNet) [14], the Chromosome Network for Biodosimetry in Japan [15], the European Network for Biological and Retrospective Physical Dosimetry (RENEB) [16] and the Biological Dose Network in China [13]. In addition to these regional networks, global networks have been created by the World Health Organization (BioDoseNet) [17], the IAEA (Response and Assistance Network, RANET), EURADOS and the Global Health Security Initiative (GHSI) [13]. Since 2018, Kazakhstan has been a member of the ARADOS (Asian Network of Biological Dosimetry) network [18], which is a platform for radiation dosimetry research among Asian countries. In addition, Kazakhstan has participated in many epidemiological studies using biological dosimetry techniques such as dicentrics, FISH translocations and physical dosimetry methods [19,20,21,22,23] as well as research on the biological effects of radiation on the human body, which further confirms the importance of research in this direction [24].

Each biodosimetry laboratory, in order to assess the individual radiation absorbed dose, is required to carried out appropriate dose–response calibration curves. To do it, an important prerequisite is to determine the level of the background frequency of dicentrics from healthy people who have not been exposed. Therefore, the starting point in the dose assessment of irradiated people is the determination of the background level of dicentrics, since the baseline level of spontaneous dicentrics can vary significantly in different populations [25,26,27].

An accurate analysis of the background frequency of dicentrics is significant, because this information is directly related to the dose estimation accuracy, especially when people are suspected of having been exposed to a low dose of radiation [24]. In the majority of studies using dicentrics analysis for occupational, accidental or medical exposure, the frequency of dicentrics before exposure (background) is not available. Since the initial “zero” dose point has a large percentage of uncertainty and becomes a variable parameter, a prerequisite is the determination of the standard background dicentric frequency. Although there is significant variability in these data in different studies, the overall average frequency of dicentrics is 1 per 1000 analyzed cells.

This value is accepted as a norm in the IAEA recommendations, although according to the data from numerous studies, it can fluctuate within fairly wide limits. Thus, in various articles devoted to the comparison of background levels of chromosome damage detected by different cytogenetic laboratories, the average frequencies of dicentrics (per 1000 cells) varied from 0.30 to 2.99 [28,29,30,31].

The implementation of studies on Kazakhstan’s background frequency of dicentrics will allow for avoiding uncertainties, negative estimates of background values and negative linear coefficients for constructing a robust and reliable calibration dose–effect curve. Therefore, the determination of the regional background frequency will allow, taking into account the above, to reasonably estimate the value of the “real” individual absorbed dose.

In the present study, an attempt to estimate the general background frequency of dicentrics and other unstable chromosome aberrations in adult individuals from different regions of Kazakhstan was carried out.

## 2. Material and Methods

### 2.1. Selection of Subjects

In this study, a group of 40 subjects (10 for each region) was selected ranked by age and sex in a 1:1 ratio. The heterogeneity of the composition of the groups in age ranges, heterogeneous ethnic composition, the presence of some bad habits, etc., justified the creation of a diverse miniature model of the region, reflecting the variability of representatives of the geographical area. As a result of the expedition, a group of 10 volunteers was formed in each region, ranked for representativeness by age characteristics in the ranges of 20–29 years, 30–39 years, 40–49 years, 50–59 years and 60–69 years and by gender. Donor groups were conditionally divided according to the geographical principle of four main directions: north, south, west and east. On the basis of the largest number and ethnic diversity of the population, cities of regional significance were selected: Petropavlovsk for the north region, Shymkent for the south region, Aktobe for the west region and Ust-Kamenogorsk for the east region. The unconditional criterion for the selection of volunteers was birth and residence in the selected region, the rural or urban history of residence was not taken into account. The selected subjects were conditionally healthy people who did not have any harmful occupational factors and had not been exposed to ionizing radiation, including medical X-ray procedures, in the last 6 months.

All subjects provided written informed consent in accordance with the current law of the Republic of Kazakhstan. All subjects involved received a paper informing them about the aims of study. They were personally interviewed by filling out one questionnaire by Carrano and Natarajan [32] for the evaluation of “lifestyle confounding factors”, except with some modifications. Subjects were questioned by passport data, place of birth and anamnesis of residence in the selected region, chronic pathologies, increased or chronic exposure to factors that can potentially affect the frequency of unstable chromosome aberrations (i.e., tobacco, alcohol, chemotherapy, radiotherapy and occupational exposure).

A gendered division of the subjects was made in equal parts (5 males and females for each region). In total, only 8 smoker subjects (20%) were included in the study. Among them, only 2 were heavy smokers (i.e., 15–20 cigarettes per day), while the others smoke approximately had from 7 to 10 cigarettes per day.

### 2.2. Whole Blood Cell Cultures and Automated Scoring

Peripheral venous blood (6 mL/subject) was taken under aseptic conditions into vacuum containers containing lithium heparin.

The cytogenetic protocol was performed according to the dicentric assay (DCA), which is currently recommended by the IAEA as the “gold standard” in biological dosimetry. Some modifications were made: colcemid was replaced by colchicin using the same proportions. The staining of slides was conducted using the standard Giemsa stain method.

An analysis of the metaphase cells was performed with an automated electron microscope AxioImager Z2 (Carl Zeiss, Jena, Germany), Metafer 4 Software (No. 2250047), MSearch automated metaphase search and photo registration system (MetaSystems, Altlußheim, Germany). ICAROS and ISIS software were used for the analysis of chromosome aberrations (MetaSystems Software).

The identification of chromosome aberrations was carried out at 1000× magnification, according to the nomenclature of chromosomes, while counting chromosomal aberrations was performed only in the cells containing 45–46 chromosomes. The dicentric chromosomes, centric rings (CRs) and excess acentric fragments were scored.

### 2.3. Statistical Analysis

The variability among donors was examined by means of the chi-square test for Poisson homogeneity and Shapiro–Wilk’s test for normal distribution. The Mann–Whitney U test and Kruskal–Wallis nonparametric test were used to evaluate differences related to chromosome aberration frequencies between the groups. Data were analyzed using the STATSOFT software.

## 3. Results

The results obtained from the chromosome aberration analysis in the group of 40 Kazakhstan control subjects are shown in Table 1. As far as the aberration analysis, a total of 168,362 cells were scored (a mean of 4209 cells per subject). In total, 133 dicentrics and centric rings and 219 excess acentric fragments were found. Considering all 40 individuals analyzed, the mean frequencies of Dic + CR of 0.84 (±0.83), Ace 1.42 (±1.31) and total 2.26 (±2.07) per 1000 cells were found (Table 1).

Shapiro–Wilk’s test showed a significant difference from the normal distribution for Dic + CR (*p* = 0.00011), Ace (*p* = 0.00004) and total chromosome aberrations (*p* = 0.00008). The frequencies of Dic + CR and Ace with good convergence were described by the Poisson distribution (chi-square test = 0.56, df = 1, *p* = 0.453 for Dic + CR; chi-square test = 3684, df = 2, *p* = 0.158 for Ace).

The frequencies of chromosome aberrations in smokers (0.87 ± 0.27 for Dic + CR; 1.27 ± 0.35 for Ace; 2.16 ± 0.62 for total) and non-smokers (0.84 ± 0.16 for Dic + CR;1.55 ± 0.22 for Ace; 2.39 ± 0.35 for total) did not show any significant differences between the two groups. Similarly, few differences in chromosome aberration frequencies between females (0.82 ± 0.16 for Dic + CR; 1.40 ± 0.29 for Ace; 2.21 ± 0.43 for total) and males (0.83 ± 0.20 for Dic + CR; 1.10 ± 0.20 for Ace; 1.94 ± 0.37 for total) were found.

The distribution of chromosome aberrations observed in Kazakhstan subjects living in different regions is shown in Table 2 and Figure 1.

The Kruskal–Wallis test showed significant differences in the frequency of all kinds of chromosome-type aberrations between the northern and western regions of Kazakhstan (Dic + CR: *p* = 0.0295; Ace: *p* = 0.024; total: *p* = 0.0104). The subjects living in the western region showed higher frequencies of chromosome aberrations also if compared with the south and east individuals, even if not significantly.

## 4. Discussion

The Republic of Kazakhstan occupies a very large territory with different levels of radiation background due to the presence of several reasons. A significant part of the territory is contaminated with natural radionuclides due to the specific geological structure. Moreover, Kazakhstan is considered to be the world’s largest producer of uranium, thorium and other minerals. In addition, it was heavily affected by a series of nuclear tests performed on the territory of the Semipalatinsk Test site (STS) in the eastern region for testing nuclear weapons for the Soviet army during the period 1949–1989. Considering this complex radiological situation, studies on Kazakhstan’s population represent important items in terms of radiological safety.

In this context, the assessment of the radiation-induced chromosome aberrations has been performed in several studies to evaluate the level of the current radiological contamination.

Regarding this, the assessment of the background frequency of chromosome unstable aberrations in the Kazakhstan population is considered of great importance.

The aim of this study was to estimate the background frequency of the radiation-induced chromosome aberrations in the Kazakhstan population by analyzing a group of control subjects living in four geographical regions with different radiological situations.

The present investigation represents the largest study related to radiation-induced chromosome damage in Kazakhstan control subjects in terms of the number of analyzed cells.

In the present study, considering entirely the studied cohort, a mean frequency of Dic + CR of 0.84 (±0.83) per 1000 cells was found. This value is very similar to other published data on Kazakhstan control subjects reported in previous studies (Table 3) [19,22,23,26,27,33]. In particular, a mean dicentric frequency of 0.71‰ was found in a group of control subjects from Kurchatov in northeast Kazakhstan [19], while a dicentric frequency of 0.78‰ was found in a group of controls from Kokpekty in East Kazakhstan [27].

The mean background frequency of dicentrics obtained in the present study were also compared to other control values reported for other populations worldwide (Table 4) [28,30,34,35,36,37,38,39,40]. These values comprised between 0.54 and 2.99/1000 cells.

A very large review reported a general dicentric background frequency of 0.78 × 10^−3^ per cell, calculated as the mean yield of observed dicentrics in more than 50 published papers on the general populations from different countries, comprising a total of 2000 healthy adult donors and 211,661 cells analyzed [35]. However, due to the significant variability among some laboratories, the authors calculated a more representative value of 0.55 × 10^−3^ for the background level of dicentrics, not considering the extreme data. Another paper reported data on the background frequencies of dicentrics from cohorts with more than 20 donors (318,636 cells analyzed from 7331 samples) from different countries [37], obtaining an overall mean dicentric frequency of 1.3 × 10^−3^/cell. In another study on a large USA control group, including 493 subjects (age range from 1.1 to 83.7 years) and a total of 108,950 cells analyzed, a mean dicentric frequency of 1.6 × 10^−3^ and a ring frequency of 0.2‰ were obtained [41].

Among the reported studies (Table 4), the highest background yield of dicentrics was 2.99 × 10^−3^ obtained from the analysis of a total of 11,700 cells from 117 people from Calcutta, India [29,38].

In the present study, the obtained results of the background frequency of radiation-induced dicentrics found in subjects living in different regions of Kazakhstan did not show strong differences compared to the literature data from general populations, except for some studies on the baseline frequencies of aberrations in countries with an increased background of natural radiation.

Comparing the frequencies of chromosome aberrations found in subjects living in four different regions in Kazakhstan, we found the highest yield of chromosomal aberrations in the West Kazakhstan region. These values were statistically different only when compared to the northern subjects.

The literature data report that some extractive industries, such as mining and oil and gas production, present in western Kazakhstan could potentially increase the natural radiation background by concentrating the amount of natural radiation above normal background levels [42,43,44,45,46,47]. Some studies performed in western Kazakhstan have shown that the region is characterized by the highest average annual effective external dose compared to the national average. This is due to the fact that several radiation hazardous facilities of various types have historically functioned in the West Kazakhstan region: the Mangyshlak Nuclear Power Plant with BN-350 reactor installations; the site of six underground nuclear explosions on the Ustyurt Plateau; the Koshkar-Ata tailings storage facility, where liquid radioactive waste from a chemical and hydrometallurgical plant and oil-bearing formations that contain radionuclides of the radioactive series U-238 and Th-232 [48,49,50] were discharged.

These circumstances may be an indirect cause of the observed significantly high frequency of spontaneous unstable chromosomal aberrations in the western region subjects of Kazakhstan.

## 5. Conclusions

In conclusion, the background frequency of chromosome aberrations found in residents from different regions of Kazakhstan were characterized by rather low-frequency values that were comparable with the data published in the literature on general populations.

The obtained data on the average frequency of the standard background level of chromosomal aberrations in the regions of Kazakhstan should be taken into account when constructing the dose–effect calibration curve as a “zero” dose point, which will reduce the uncertainty in the quantitative assessment of the individual absorbed dose in the case of emergency radiological situations.

## Figures and Tables

**Figure 1 ijerph-19-08485-f001:**
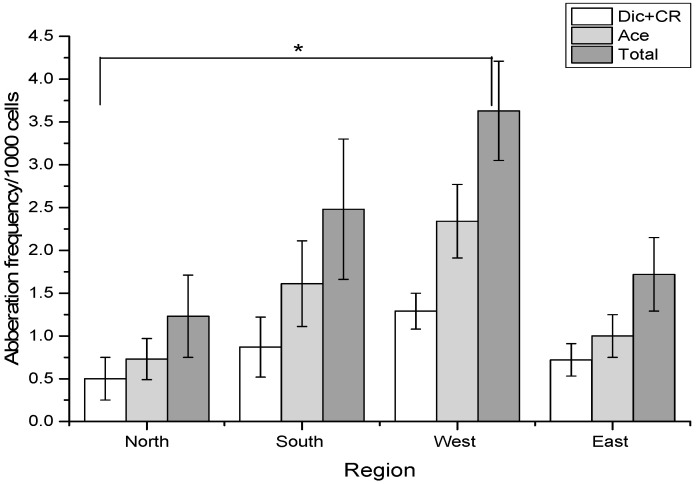
Chromosome aberration frequencies in different regions of Kazakhstan. The asterisk indicates a significant difference (*p* < 0.05) between Dic + CR, Ace and total chromosome aberrations between the northern and western groups.

**Table 1 ijerph-19-08485-t001:** Frequency of chromosome aberrations in control subjects from Kazakhstan regions.

№	Subject	Age	Sex	Cells Scored	Dic + CR	Dic + CR/1000 Cell	Ace	Ace/1000 Cell	Total	Total/1000 Cell
	**North Kazakhstan, Petropavlovsk**									
1	N1	27	M	4951	1	0.20	1	0.20	2	0.40
2	N2	33	M	5155	1	0.19	2	0.39	3	0.58
3	N3	44	M	6231	0	0.00	1	0.16	1	0.16
4	N4	55	M	5569	1	0.18	3	0.54	4	0.72
5	N5	61	M	6800	6	0.88	9	1.32	15	2.21
6	N6	28	F	2130	0	0.00	1	0.47	1	0.47
7	N7	36	F	5369	0	0.00	1	0.19	1	0.19
8	N8	49	F	6904	0	0.00	3	0.43	3	0.43
9	N9	55	F	7962	9	1.13	8	1.00	17	2.14
10	N10	64	F	6156	15	2.44	16	2.60	31	5.04
	Total			57,227	33	0.58	45	0.79	78	1.36
	Mean			5723	6	0.50	5	0.73	8	1.23
	±SD			1563	6	0.79	5	0.75	10	1.53
	SE			494	2	0.25	2	0.24	3	0.48
	**South Kazakhstan, Shymkent**									
11	S1	27	M	4309	2	0.46	2	0.46	4	0.93
12	S2	34	M	3010	1	0.33	4	1.33	5	1.66
13	S3	47	M	4160	0	0.00	11	2.64	11	2.64
14	S4	58	M	3168	4	1.26	4	1.26	8	2.53
15	S5	68	M	2369	9	3.80	13	5.49	22	9.29
16	S6	28	F	3197	1	0.31	1	0.31	2	0.63
17	S7	31	F	3621	1	0.28	3	0.83	4	1.10
18	S8	43	F	2404	0	0.00	1	0.42	1	0.42
19	S9	53	F	3376	6	1.78	10	2.96	16	4.74
20	S10	60	F	4594	2	0.44	2	0.44	4	0.87
	Total			34,208	26	0.76	51	1.49	77	2.25
	Mean			3421	3	0.87	5	1.61	8	2.48
	±SD			758	3	1.11	4	1.57	7	2.58
	SE			240	1	0.35	1	0.50	2	0.82
	**West Kazakhstan, Aktobe**									
21	W1	27	M	3135	4	1.28	6	1.91	10	3.19
22	W2	33	M	2939	4	1.36	4	1.36	8	2.72
23	W3	46	M	3427	1	0.29	1	0.29	2	0.58
24	W4	56	M	4745	2	0.42	5	1.05	7	1.48
25	W5	64	M	3917	9	2.30	10	2.55	19	4.85
26	W6	29	F	3611	4	1.11	15	4.15	19	5.26
27	W7	35	F	3024	6	1.98	9	2.98	15	4.96
28	W8	49	F	3795	7	1.84	18	4.74	25	6.59
29	W9	52	F	3578	3	0.84	7	1.96	10	2.79
30	W10	62	F	3377	5	1.48	8	2.37	13	3.85
	Total			35,548	45	1.27	83	2.33	128	3.60
	Mean			3555	5	1.29	8	2.34	13	3.63
	±SD			526	2	0.65	5	1.36	7	1.84
	SE			166	1	0.21	2	0.43	2	0.58
	**East Kazakhstan, Ust-Kamenogorsk**									
31	E1	27	M	4031	1	0.25	3	0.74	4	0.99
32	E2	31	M	2617	4	1.53	6	2.29	10	3.82
33	E3	46	M	3000	1	0.33	1	0.33	2	0.67
34	E4	55	M	4346	4	0.92	6	1.38	10	2.30
35	E5	62	M	4650	3	0.65	3	0.65	6	1.29
36	E6	21	F	4981	1	0.20	2	0.40	3	0.60
37	E7	38	F	3534	1	0.28	1	0.28	2	0.57
38	E8	45	F	3864	2	0.52	4	1.04	6	1.55
39	E9	50	F	5767	3	0.52	3	0.52	6	1.04
40	E10	64	F	4589	9	1.96	11	2.40	20	4.36
	Total			41,379	29	0.70	40	0.97	69	1.67
	Mean			4138	3	0.72	4	1.00	7	1.72
	±SD			939	2	0.59	3	0.78	5	1.36
	SE			297	1	0.19	1	0.25	2	0.43
**Total**										
	Total			168,362	133	0.79	219	1.30	352	2.09
	Mean			4209.05	3.91	0.84	5.48	1.42	8.80	2.26
	±SD			1349.74	3.31	0.85	4.62	1.31	7.55	2.07
	SE			213.41	0.52	0.13	0.73	0.21	1.19	0.33

Dic: dicentric; CR: centric ring; Ace: excess acentric fragment.

**Table 2 ijerph-19-08485-t002:** Chromosome aberration yield in subjects living in different regions of Kazakhstan.

Region	Subjects	Mean Age ± SD	Chromosome Aberrations Per 1000 Cells ± SE		
			Dic + CR	Ace	Total
North Kazakhstan	10	45 ± 12	0.50 ± 0.25	0.73 ± 0.24	1.23 ±0.48
South Kazakhstan	10	45 ± 14	0.87 ± 0.35	1.61 ± 0.20	2.48 ± 0.82
West Kazakhstan	10	45 ± 13	*** 1.29 ± 0.21**	*** 2.34 ± 0.43**	*** 3.63 ± 0.58**
East Kazakhstan	10	44 ± 14	0.72 ± 0.19	1.00 ± 0.25	1.72 ± 0.43

SD: Standard deviation; SE: standard error; Dic: dicentric; CR: centric ring; Ace: excess acentric fragment. * Statistically significant (*p* < 0.05).

**Table 3 ijerph-19-08485-t003:** Comparison of chromosome aberration frequencies in the Kazakhstan control groups.

Reference	Number of Subjects	Number of Cells	Dic + CR	Dic + CR(‰)	Ace	Ace (‰)
Testa et al., 2001	**20**	4000	3	0.71 (±0.18)	22	5.2 (±0.91)
Svyatova et al., 2002	**25**	8697	4	0.46 (±NR)	17	2.0 (±0.07)
Abil’dinova et al., 2003	**25**	8716	2	0.23 (±NR)	17	2.0 (±0.1)
Tanaka et al., 2006	**46**	14,192	11	0.78 (±2.2)	NR	NR
Takeichi et al., 2006	**18**	6600	1	0.2 (±NR)	7	1.1 (±NR)
Djansugurova et al., 2020	**236**	22,642	NR	0.13 (±0.091)	NR	NR
This paper	**40**	168,362	133	0.84 (±0.83)	219	1.42 (±1.31)

Dic: dicentric; CR: centric ring; Ace: excess acentric fragment; NR: not reported.

**Table 4 ijerph-19-08485-t004:** Chromosome aberration background frequencies in control subjects from different populations.

Reference	Country (City, Village)	Number of Subjects	Number of Cells	Dic	CR	Dic + CR	Dic/Dic + CR (‰)
Barcinski et al., 1975	Brazil (Saquarema)	147	9001	6	0	6	0.67
Lloyd et al., 1980	Review of published data from different countries	2000	211,661	166	NR	166	0.78
Gundy et al., 1983	Hungary	175	17,500	12	0	12	0.69
Bender et al., 1988	Review of published data from different countries	7331	318,636	430	48	478	1.3
Ganguly et al., 1993	India (Calcutta)	117	11,700	35	NR	35	2.99
Bauchinger et al., 1994	Germany (Munich)	85	45,952	19	6	25	0.54
Stephan et al., 1999	Germany	53	54,689	63	4	67	1.23
Santovito et al., 2015	Italy (Turin)	101	20,200	11	28	39	1.93
Karuppasamy et al., 2018	India (Kerala)	97	25,359	36	3	39	1.54
This paper	Kazakhstan	40	168,362	101	32	133	0.84

Dic: dicentric; CR: centric ring; NR: not reported.

## Data Availability

Detailed information confirming the published results can be found in the archived data sets of the National Center of Science and Technology Evaluation the Republic of Kazakhstan, report number No. 0219RK00797.

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
