# Peer review of "Background Level of Unstable Chromosome Aberrations in the Kazakhstan Population: A Human Biomonitoring Study"

_ijerph, 2022, doi:10.3390/ijerph19148485_

Round 1
Reviewer 1 Report
Review of paper “Background Level of Unstable Chromosome Aberrations in the 2 Kazakhstan Population: A Human Biomonitoring Study” by L. Kenzhina et al.
General comments
The paper reports the background level of selected chromosome aberrations (dicentrics and acentrics) in 4 different regions of Kazakhstan. This could be useful for cytogenetic biodosimetry study, as a zero dose point is needed.
The paper looks fine to me. I don’t have many comments apart from several typological errors noted.
The number of subjects (40) is rather low compared to similar studies presented in Table 4.
I am not familiar with the experimental technique reported. This part will have to be assessed by another reviewer.
Specific comments
Introduction, line 40: “like no other countries in the world.” I would remove that. Japan has experienced 2 nuclear bombs. Many epidemiological studies are based on atomic bomb survivors.
Line 52: what are ore and thorium provinces ?
Author Response
The Authors thank the Editor for the congratulations about the manuscript and the Reviewers for their appropriate and valuable comments.
The following changes have been made to the manuscript according to the Reviewers' comments.
Reviewer 1
Line 41: the sentence “like no other countries in the world” was removed, as suggested by the Reviewer.
Lines 54-55: information about uranium ore and thorium provinces were added

Reviewer 2 Report
Authors checked in 40 persons (all smokers obviously) for chromosomal aberrations in peripheral blood and did not find enhanced rates compared to normal controls from other countries, even though Kazakhstan is a country with high radioactive background irradiation.
Results are interesting – still as in Soviet Union many similar studies have been done authors are kindly ask to refer to corresponding historical Russian papers as well.
Also minor comments:
- Abbreviations dic, cr, ace, need to be explained in text at first use
- In Tab 1 please replace ‘,’ by ‘.’
Author Response
The Authors thank the Editor for the congratulations about the manuscript and the Reviewers for their appropriate and valuable comments.
The following changes have been made to the manuscript according to the Reviewers' comments.
Reviewer 2
We specify that among the 40 subjects recruited in the study only 8 smokers were included, as reported in the Materials and Methods (line 142).
The abbreviations were checked and explained at the first use in the text (Lines 62-63, line 159)
In Table 1 and Table 2 the “,” were replaced by “.”

Reviewer 3 Report
The manuscript presented the background frequency of unstable chromosome aberrations in the population of Kazakhstan. The study is clearly written, has the appropriate references and background information. The overall quality of the manuscript is good.
1- Authors selected 10 donors per city and grouped by age and gender (1:1). Could you provide more info on the sampling methods of 10 subjects, and how you decided to use just 10 donors from each area in North, South, West, East Kazakhstan, and to cover a wide rage of age (20-70).
2- Pls provide some numeric results in the Abstract when modify the manuscript;
Author Response
The Authors thank the Editor for the congratulations about the manuscript and the Reviewers for their appropriate and valuable comments.
The following changes have been made to the manuscript according to the Reviewers' comments.
Reviewer 3
- Line 117-122: information on the sampling methods of the study subjects was added
- Line 24-26: the numeric results on the background frequencies found in this study and the general background frequencies reported in literature were added in the Abstract.
